# A Multisample Approach in Forensic Phenotyping of Chronological Old Skeletal Remains Using Massive Parallel Sequencing (MPS) Technology

**DOI:** 10.3390/genes14071449

**Published:** 2023-07-14

**Authors:** Jezerka Inkret, Tomaž Zupanc, Irena Zupanič Pajnič

**Affiliations:** Institute of Forensic Medicine, Faculty of Medicine, University of Ljubljana, Korytkova 2, 1000 Ljubljana, Slovenia; jezerka.inkret@mf.uni-lj.si (J.I.); tomaz.zupanc@mf.uni-lj.si (T.Z.)

**Keywords:** skeletal remains, forensic DNA phenotyping, eye color, hair color, HIrisPlex, Second World War

## Abstract

It is very important to generate phenotypic results that are reliable when processing chronological old skeletal remains for cases involving the identification of missing persons. To improve the success of pigmentation prediction in Second World War victims, three bones from each of the eight skeletons analyzed were included in the study, which makes it possible to generate a consensus profile. The PowerQuant System was used for quantification, the ESI 17 Fast System was used for STR typing, and a customized version of the HIrisPlex panel was used for PCR-MPS. The HID Ion Chef Instrument was used for library preparation and templating. Sequencing was performed with the Ion GeneStudio S5 System. Identical full profiles and identical hair and eye color predictions were achieved from three bones analyzed per skeleton. Blue eye color was predicted in five skeletons and brown in three skeletons. Blond hair color was predicted in one skeleton, blond to dark blond in three skeletons, brown to dark brown in two skeletons, and dark brown to black in two skeletons. The reproducibility and reliability of the results proved the multisample analysis method to be beneficial for phenotyping chronological old skeletons because differences in DNA yields in different bone types provide a greater possibility of obtaining a better-quality consensus profile.

## 1. Introduction

Externally visible characteristics (EVCs) are complex phenotypic traits that have been gaining exceptional value in recent genetic studies, especially in the forensic context [1,2,3,4]. They make it possible to obtain useful information about what the deceased person may have looked like, such as eye, hair, and skin color [5,6,7]. In addition to EVC, it is possible to obtain biogeographical ancestry information from biological material [8,9,10,11,12]. Prediction of external features and biogeographical origin can provide police investigators with more leads if standard DNA profiling is not informative or if there are no comparative samples for solving missing person identification cases or investigating biological traces in crime cases. For eye, hair, and skin color prediction, the HIrisPlex and HIrisPlex-S systems supported by large genotype and phenotype databases are used [13]. Several studies have applied HIrisPlex and HIrisPlex-S systems to samples extracted from chronological old and even ancient bones and teeth [14,15], and some historical questions have been answered through comparison of EVC prediction and historical documentation [1,16,17,18,19].

It is important to generate phenotypic results that are reliable when processing chronological old skeletal remains for cases involving the identification of missing persons. In particular, the preservation of DNA is affected not only by the environment in which human remains are found but also by the time when exposure to unfavorable environmental conditions took place [20,21,22]. The DNA that is extracted from chronological old bones is generally poorly preserved, and the quantity and quality of the DNA are low [23]. Poor preservation of DNA was considered in the instructions for disaster victim identification (DVI), and according to Prinz et al. [24], investigations need to be duplicated on the same sample (recommendation number 7) and, if possible, samples from a larger number of skeletal elements of one skeleton should be taken at the earliest possible stage of investigation (recommendation number 3). The application of PCR-MPS techniques [25,26] allows EVC kits to be developed [27,28]. The advantage of MPS technology compared to SNaPshot technology followed by capillary electrophoresis is a lower DNA input needed for running one assay for many different single nucleotide polymorphisms (SNPs). This consequently reduces the amount of DNA needed for analysis, the efforts of laboratory personnel, and ultimately time [27,29]. The disadvantage of using PCR-MPS reagents is the high cost, and some published studies on phenotyping prediction of chronological old bones have been performed using a single test with no repetition. The results of the single-test approach applied to bones up to 80 years old showed that only half of the samples analyzed produced a full HIrisPlex-S profile due to low bone DNA quantity [14]. Even if an optimal amount of template DNA (1 ng) was used for the PCR MPS HIrisPlex reaction, a single-test approach with no repetition resulted in different levels of AUC (area under the receiver operating curve) loss in 4.2% of the HIrisPlex markers [15], indicating the need for at least replicate analyses of the same sample to achieve reliable MPS phenotyping results. However, following the third DVI recommendation [24], multiple samples should be collected for analysis, and it is well known that different skeletal element types that belong to different anatomical regions contain different amounts of DNA [30,31,32], which in principle offers a greater chance of obtaining a higher-quality profile than if a single bone is used and investigations are duplicated. Gray (1918) [33] classified human bones as short, long, flat, irregular, or sesamoid, and bones also vary by shape and function. Consequently, bone composition also varies because some bones have thicker tissue that is more compact with greater mineralization, and other bones are mostly comprised of trabecular bone covered by a thin to medium-thick layer of compact bone. Long bones are predominantly thick compact bone with a limited share of cancellous bone, whereas short, sesamoid, flat, and irregular bones primarily have cancellous bone in comparison to compact bone tissue (Gray 1918) [33]. To improve the success of pigmentation prediction in chronological old skeletal remains, a multisample approach was tested on eight skeletons using PCR-MPS technology. Three different bone types were analyzed per skeleton, which makes it possible to obtain a consensus profile, thereby increasing the reproducibility and reliability of the prediction. Obtaining a large collection of skeletons from forensic cases is not possible, and so skeletons from victims of the Second World War were used as proxies for badly preserved skeletal remains.

## 2. Material and Methods

### 2.1. Sample Selection and DNA Extraction

The samples were selected from the Huda Jama Mass Grave (Slovenia), in which 432 victims of a mass killing were recovered [34]. Eight skeletons were used for HIrisPlex MPS analysis and three bone types were selected per skeleton. Based on the completeness and state of preservation of the skeletal remains, bones that yield the highest amounts of DNA according to previous studies [32,34,35] were selected. Appendix A presents the inventory of bones used. Mechanical and chemical cleaning of bones was carried out to remove surface contamination. Because the work involved skeletal elements with a long postmortem interval (PMI), special procedures were followed to avoid contamination of the specimens with modern DNA [21]. DNA was extracted from half a gram of bone powder following the procedure by Zupanič Pajnič [36]. Purification was performed according to instructions provided by the manufacturer with a Biorobot EZ1 device (Qiagen, Hilden, Germany) and the EZ1 DNA Investigator Kit (Qiagen). DNA was eluted in 50 μL of TE buffer. Following Parson et al. [37], ENCs were included to monitor possible contamination during the extraction procedure and to ensure that the extraction plastics and reagents were clean. DNA was also isolated from buccal swabs of personnel participating and handling samples to create an elimination database. For DNA extraction from buccal swabs, the EZ1 DNA Investigator Kit (Qiagen) was used following the manufacturer’s instructions [38].

### 2.2. DNA Quantification

The DNA quantity and quality were measured with the PowerQuant System (Promega, Madison, WI, USA) following the technical manual [39]. The degree of DNA degradation (degradation index, DI) was calculated according to the ratio between short autosomal (Auto) and long degradation (Deg) targets. The DNA concentration of the short target was used for PCR DNA input calculation. The presence of possible PCR inhibitors was determined through internal positive control (IPC). Quantification data were obtained with the ABI 7500 Real-Time PCR System (Applied Biosystems, Foster City, CA, USA) and PowerQuant Analysis Tool software—Promega (https://worldwide.promega.com/resources/tools/powerquant-analysis-tool, accessed on 19 May 2022). The positive and negative controls were carried out in duplicate amplifications along with the bone extracts. Negative template controls and ENCs were analyzed to verify the cleanliness of laboratory plastics and reagents.

### 2.3. STR Typing

STR typing was conducted to confirm that the three different skeletal elements selected from each skeleton indeed had the same genetic profile for identity confirmation. To exclude any contamination during the extraction process, ENCs were investigated for STRs (HIrisPlex analysis of ENCs was not performed due to the high cost of the MPS reagents). To obtain autosomal genetic profiles, the PowerPlex ESI 17 Fast System (Promega) was used based on the manufacturer’s protocols [40]. One ng of DNA served as a template when the quantification of the short PowerQuant target was 0.058 ng or more per μL of extract. When the quantification was lower, the maximum volume (17.5 μL) of DNA extract was used. Amplification was performed in the Nexus Master Cycler (Eppendorf, Hamburg, Germany). In addition to skeletal samples, positive and negative controls were also amplified. For the amplification of ENCs, the maximum volume of extracts was used. The genetic profiles were obtained using the SeqStudio Genetic Analyzer for HID (Thermo Fisher Scientific, TFS, Carlsbad, CA, USA) together with the WEN Internal Lane Standard 500 (Promega), SeqStudio Data Collection Software v 1.2.1 (TFS), and GeneMapper ID-X Software v. 1.6 (TFS). STR typing was also performed for persons in the elimination database with the NGM kit [41]. The genetic profiles in the elimination database were compared to those acquired from bones to control for potential contamination of aged DNA by contemporary DNA.

### 2.4. HIrisPlex SNP Typing

Hair and eye color predictions were made with a customized version of the PCR-MPS HIrisPlex panel [14,27] and 24 SNPs were analyzed on the Ion S5 System MPS platform (TFS). For library preparation, 1 ng of DNA was used for 14 samples (Appendix A, labeled in bold), as recommended by the manufacturer [42]. Because 15 μL of DNA can be used as a maximum volume for library preparation, samples with a concentration higher than 0.067 ng per μL reached the amount of 1 ng DNA input for the PCR-MPS HIrisPlex reaction. For the other 10 samples, 0.15 to 0.9 ng of DNA was used as a template (Appendix A, column DNA input quantity). PCR with target amplification was run at the recommended number of 25 cycles for samples with an optimal amount of DNA input and for two samples for which 0.6 and 0.9 ng of DNA input were used. Eight samples with an optimal amount of DNA were combined in library pool 1, six samples with 1 ng of DNA input and two samples with 0.6 and 0.9 ng of DNA input were combined in library pool 2, and eight samples in which less than 0.3 ng of DNA was used as a template (Appendix A, labeled in red) were combined in library pool 3, for which the number of cycles was increased to 27. The Precision ID DL8 Kit for Ion Chef was used for library preparation following the manufacturer’s instructions [42]. The concentration of the combined library pools was determined with the Ion Library TaqMan Quantification Kit (TFS) according to the manufacturer’s recommendations [43]. The library pool was diluted to 30 pM combining all 24 samples. The Ion Chef System (TFS) was used for DNA template preparation. The template was loaded onto an Ion 530 sequencing chip (TFS) in a single chip loading workflow. For sequencing, Ion S5 Precision ID sequencing reagents (TFS) and Ion S5 Precision ID sequencing solutions (TFS) were used. The alignment of reads against the *Homo sapiens* reference genome (hg19) was performed using Ion Torrent Suit Software 5.6 (TFS). The data on the level of sequence coverage in targeted regions were obtained using the Coverage Analysis v. 5.6.0.1. Plugin, and Converge software version 2.0 (TFS) was used for genotyping. The default setting was applied based on the manufacturer’s recommendations [43].

Because three samples per skeleton were analyzed, the consensus genotype was determined as advised when multiple samples were available for the same skeleton [44,45,46,47,48] according to Turchi et al. [49]. For the HIrisPlex pigmentation prediction, the R-script (https://walshlab.sitehost.iu.edu, accessed on 12 July 2022) was used to convert the constructed consensus profiles into the corresponding input codes, which were uploaded into the HirisPlex web tool (accessible at https://hirisplex.erasmusmc.nl, accessed on 12 July 2022). Eye and hair color prediction was achieved with the most likely eye and hair color prediction according to *p*-value and AUC loss.

### 2.5. Statistical Analysis

Microsoft Excel 2016 was used for statistical calculations and the generation of graphs. To determine the variability in the efficiency of sequencing HirisPlex SNP markers between skeletons, the average coverage value obtained from three bone types was calculated per each skeleton, and average coverage values were also calculated for eight skeletons considering data obtained from all 24 bones analyzed.

## 3. Results

### 3.1. DNA Quantity and STR Typing

The average results of DNA quantity for bone samples and ENCs are shown in Figure 1 and Appendix A. The quantity of DNA obtained from 1 g of bone powder is also shown. Deg target was detected in all bones except in metatarsal II from skeleton 4, for which it was not possible to determine DI. DI varied widely (from 2.2 to 40) across the skeletons and even between different bone types (except for skeletons 1 and 7), indicating high variability in DNA preservation between the bones analyzed. The lowest degradation was observed in skeletons 1 and 7, and the highest in skeleton 5 (Appendix A). The IPC shift value was below the threshold of 0.30 in all samples analyzed, showing efficient purification of DNA. DNA quantity ranged from 1 ng/g of bone to 48 ng/g of bone, and, on average, 14 ng/g of bone was extracted across all skeletal samples tested. The optimal amount of DNA for the PCR-MPS HIrisPlex reaction was obtained from 14 bone samples (Appendix A, labeled in bold), showing that almost half of the bones did not reach 1 ng of DNA input for HIrisPlex analysis. No PowerQuant targets were amplified for ENCs.

The amount of DNA extracted varies between skeletons. The highest average amounts of DNA considering all three skeletal element types were obtained from skeletons 6, 7, and 8 (average 0.3 ng/µL), and the lowest from skeletons 1, 2, and 4 (average 0.02 ng/µL; see Figure 1). High variability in DNA content was also observed between different skeletal elements that were analyzed for each skeleton, which can be seen in Figure 1 and in greater detail in Appendix A.

The results of STR typing are summarized in Appendix A. The success of STR amplification is expressed as the number of successfully typed loci out of all loci present in the STR kit. The results show that full STR profiles were obtained from all samples analyzed except metatarsal II from skeleton 4, for which only 175 pg of DNA was amplified and 11 loci were amplified successfully (Appendix A, labeled in green). Identical STR profiles were obtained from different bone samples from the same skeleton for each of the eight skeletons. According to criteria for internally validated genetic profile submissions, samples containing a minimum of 11 STR loci plus amelogenin are eligible to be submitted to the International Commission on Missing Person database [50], and so comparison of the partial genotype of metatarsal II from skeleton 4 to the other two bones from the same skeleton was possible and a match was confirmed. However, for all skeletons, STR profiling confirmed that the three skeletal elements sampled from the same skeleton indeed belonged to the same individual. ENC samples did not produce any autosomal STR profile, indicating that no contamination occurred during the extraction process.

### 3.2. HIrisPlex SNP Typing and Prediction of Eye and Hair Color

Quantification results showed that all three library pools reached more than 30 pM of DNA (first pool 72 pM, second pool 83 pM, and third pool 74 pM). The main sequencing parameters (average mapped reads, percentage of on-target reads, mean depth, and percentage of uniformity of sequencing) of the samples analyzed are shown in Appendix A. The libraries yielded 86.3 to 98.6% of on-target reads, and only two bone samples yielded less than 90% of on-target reads (Appendix A, labeled in green). One hundred percent uniformity of sequencing was mostly reached in skeletons with bones that yielded enough DNA for 1 ng of template input for the HIrisPlex reaction (skeletons 5, 6, 7, and 8). When less than 1 ng of DNA input was used, the uniformity of sequencing was lower, but no less than 92.4%. However, in four samples, 100% uniformity of sequencing was reached even if less than 1 ng of template input was used (Appendix A, labeled in red).

All 24 libraries produced reads above the 20× threshold (settings from Converge software, version 2.0, TFS). The allelic imbalance was flagged by Converge software only in one heterozygous genotype (sample S9, rs 1393350), and one homozygous genotype was flagged for the presence of low covered nucleotide that was below the threshold (sample S6, rs 2228479). For both samples, an increased number of cycles was used for amplification, which can lead to artifacts; that is, drop-in events and allelic imbalances. However, in both samples, genotypes were determined by Converge software as heterozygous for sample S9 and as homozygous for sample S6.

The coverage values for each HIrisPlex marker for all samples analyzed are shown in Appendix A. Together with the coverage values, the threshold for >100 pg DNA input is shown. The thresholds differ between SNP markers in line with the validation study of the HIrisPlex-S panel [27]. In all 24 bone samples analyzed, the coverage values are at significantly higher values than the recommended thresholds, which is consistent with the fact that 1 ng of DNA was amplified in more than half of the samples and that for samples in which DNA input was lower than 0.3 ng, an increased number of cycles was used. In Appendix A, coverage values for 24 SNPs are shown for three different skeletal elements per skeleton. Because 1 ng of DNA was used as DNA input in all three skeletal element types analyzed for a particular skeleton only in four skeletons, comparison between different skeletal elements and coverage values was possible only for skeletons 5, 6, 7, and 8. In skeleton 6 (metacarpal II, metatarsal II, and the capitate bone were analyzed) and in skeleton 8 (metacarpal IV, metatarsal IV, and metacarpal V were analyzed) the coverage values were very similar for all three skeletal element types, whereas differences were observed in skeletons 5 and 7. In skeleton 5, coverage was similar for metacarpal IV and metacarpal II, whereas lower coverage was observed for the thoracic vertebra. In skeleton 7, coverage was the highest for metatarsal I and much lower but very similar for the medial cuneiform and first proximal hand phalanx.

When the average coverage values calculated for each skeleton were compared between SNP markers, we noticed that lower coverage was achieved for six markers (rs 1805005, 1805006, 2228479, 16891982, 12821256, and 4959270) regardless of the skeleton analyzed (Appendix A and Appendix A). The length of the first three SNP markers mentioned is about 110 bp, and the length of the last three markers is from 45 to 70 bp. The same was true when the average coverage was calculated for all skeletons analyzed, and the same markers yielded the lowest coverage. The results obtained on our sample set showed that the sequencing efficiency of different HIrisPlex markers was not related to the length of the markers analyzed (Appendix A).

All 24 samples generated complete HIrisPlex profiles, and the results are shown in Appendix A. Skeletal samples that were already confirmed with STR typing to be from the same individual also showed identical HIrisPlex genotypes. Consensus profiles were determined for each skeleton for all SNPs analyzed, and Appendix A show the consensus profiles generated for all eight skeletons. Because HIrisPlex profiles matched between three bone types sampled per skeleton, no doubt exists regarding the accuracy of consensus profile determination. No consensus profile showed identity with any other consensus profile determined for the skeletons analyzed, indicating no cross-contamination events in the HIrisPlex MPS analysis.

Consensus profiles were used for eye and hair color prediction using the HIrisPlex web tool. Table 1 shows probability values (*p*-values) for different eye and hair colors and shades, and the most likely predicted phenotype for the skeletons analyzed. The AUC loss in all samples tested was 0 because there were no missing data for the analyses. Successful hair and eye color prediction was achieved for all skeletons. For hair color prediction, *p*-values were below 0.7 for all skeletons except skeletons 7 and 8, and for eye color above 0.9 for six skeletons. The most likely eye color phenotype was blue in five skeletons and brown in three skeletons. Blond to dark blond hair was predicted in three skeletons, brown to dark brown and dark brown to black in two skeletons, respectively, and blond in one skeleton (Table 1).

## 4. Discussion

When analyzing eight skeletons, the average amount of DNA of all three selected skeletal elements differed between skeletons and between the bones of an individual skeleton, which is in agreement with studies already published [22,30,31,32,35,51]. DNA content in different types of bones depends on the proportion between the cancellous bone tissue (more porous and flexible tissue that often contains red bone marrow) and compact bone tissue (a stronger and denser tissue) that bones are composed of. Moreover, various bones—as well as parts of identical bone—show histomorphological variability [52], and significant differences can be seen at the nanoscale for compact and cancellous bone [53]. Skeletons and bones with different DNA yields and different levels of degradation (skeletons 1 and 7 were less degraded than other skeletons) were analyzed and, when less than 0.3 ng of template DNA was used for library preparation, an increased number of PCR cycles was used. In this way, a full HIrisPlex profile with high coverage values was successfully obtained even from metatarsal II from skeleton 4, where the PowerQuant Deg target was not detected and only 0.15 ng of DNA was amplified. The approach of increasing the number of PCR cycles for two cycles when less than 0.3 ng DNA input was used proved to be very successful for phenotyping because the profiles obtained were identical to the bones amplified with the standard number of PCR cycles and higher DNA input. Only in skeleton 2 were DNA yields lower than 0.3 ng of DNA input in all three bone types analyzed, and for all of them, an increased number of cycles was used. Nevertheless, all three bones yielded an identical full HIrisPlex profile with high coverage values.

The results of this study confirm the effectiveness of a multisample approach for PCR-MPS HIrisPlex-based eye and hair color prediction with reproducible and reliable results in all bones analyzed. Differences in DNA content in different bone types [34,35] were shown to be beneficial when using a multisample approach for phenotyping chronological old skeletons because these differences provide a greater possibility of obtaining a better-quality consensus profile than when duplicated analyses of the same sample are used. The PCR-MPS HIrisPlex-S system was recently used for human pigmentation prediction in 63 skeletal samples with PMI ranging from 1 to 78 years; only one bone was tested from each skeleton, and no duplication of analysis was performed. Because of high DNA degradation, only 35 samples produced a full HIrisPlex-S profile [14]. A multisample approach, which offers the possibility of consensus profile generation, could improve phenotyping results, and bones that usually produce higher DNA yields should be selected for phenotyping [34,35]. Of course, a multisample approach cannot be used in phenotyping missing persons in cases when only an individual bone is found, nor is it useful for phenotyping commingled skeletal remains from mass graves. A recent study performed on ancient skeletons using an MPS-based HIrisPlex system also proposed a multisample analysis method with consensus profile generation as a suitable method for eye and hair color prediction of ancient DNA obtained from 11 skeletons dated from the 3rd to the 18th centuries AD [15].

When analyzing 24 bones, no contamination issue was confirmed by the clean ENCs, there was no match between the STR profiles of the bones analyzed and individuals from the elimination database, and by both STR and HIrisPlex genotype matches of three different skeletal elements sampled from eight skeletons.

The HIrisPlex system using SNaPshot technology followed by capillary electrophoresis (CE) has been applied to analysis of victims from Second World War mass graves in Slovenia (Chaitanya et al. 2017) [54], and the results have effectively demonstrated the practical usefulness of predicting phenotypic traits. Namely, two femurs from the same mass grave were found to belong to brothers based on STR typing. With the help of phenotypic prediction and anamnestic data on their appearance, based on the testimony of their sister, it was determined which brother was which (Chaitanya et al. 2017) [54]. Similar was performed by Zupanič Pajnič [19] by applying the HIrisPlex system and massive parallel sequencing (MPS) technology to a female victim of an elite prewar couple killed during the Second World War. The female victim from the grave could not be identified through autosomal STR and mitochondrial DNA analyses because she had no living relatives [19]. However, she was a prominent person of the time, and her portrait hangs in the Ljubljana City Museum, and so it was possible to compare the eye and hair color with her appearance in the portrait [19].

**Key** **points**

Phenotypic results that are reliable are of great importance for chronological old skeletons.The possibility of consensus profile generation improves the success and reliability of eye and hair color prediction.PCR-MPS HIrisPlex-based eye and hair color prediction was performed on eight Second World War victims, and three different bone types were selected from each skeleton.Full HIrisPlex profiles and full phenotypic predictions were achieved from all bones analyzed.Reproducible and reliable results showed the multisample approach to be effective for PCR-MPS HIrisPlex analyses.

## Figures and Tables

**Figure 1 genes-14-01449-f001:**
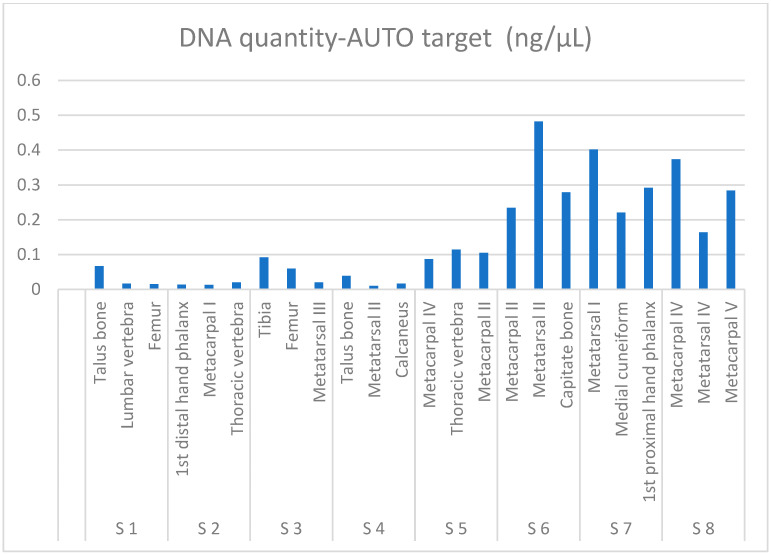
DNA quantity obtained using the qPCR PowerQuant System (Promega) expressed in ng of DNA per microliter of extract for three skeletal elements per each skeleton analyzed (S1 to S8). Results were obtained from Auto target amplification measures.

**Table 1 genes-14-01449-t001:** Eye and hair color prediction results for eight skeletons with *p*-values for eye and hair colors and shades, and the most likely predicted phenotype.

Skeleton	Hair Color and Shade Probability (*p*-Values)						Eye Color Probability (*p*-Values)			Predicted Phenotype
	Blond	Brown	Red	Black	Light	Dark	Blue	Intermediate	Brown	
1	0.658	0.309	0.004	0.029	0.952	0.048	0.911	0.057	0.032	Blue eyes Blond to dark blond hair
2	0.667	0.285	0.002	0.046	0.933	0.067	0.949	0.035	0.016	Blue eyes Blond to dark blond hair
3	0.122	0.602	0.003	0.273	0.388	0.612	0.000	0.016	0.983	Brown eyes Dark brown to black hair
4	0.38	0.548	0.007	0.085	0.858	0.142	0.926	0.057	0.017	Blue eyes Brown to dark brown hair
5	0.416	0.464	0.004	0.115	0.731	0.269	0.143	0.227	0.629	Brown eyes Dark brown to black hair
6	0.69	0.278	0.003	0.03	0.953	0.047	0.932	0.046	0.021	Blue eyes Blond to dark blond hair
7	0.218	0.744	0.006	0.032	0.779	0.221	0.05	0.114	0.836	Brown eyes Brown to dark brown hair
8	0.701	0.072	0.222	0.004	1	0	0.932	0.046	0.021	Blue eyes Blond hair

## Data Availability

The authors declare that all the data are available.

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
