# Peer review of "A Multisample Approach in Forensic Phenotyping of Chronological Old Skeletal Remains Using Massive Parallel Sequencing (MPS) Technology"

_genes, 2023, doi:10.3390/genes14071449_

Round 1

Reviewer 1 Report

The manuscript describes a succesful study of eight excavated skeletons from a mass grave. In general everything is described clearly and in detail. However, I'm rather surprised by the way the results are discussed / concluded. If I understand everything the right way, it seems like all the profiles of the pigmentation markers were complete and concordant. While this does support that the different bones came from the same skeleton it does not show much support for the multi-sample approach. 

Also, the mentioned 'consensus' profile that was used for calculation could just as well come from the same bone.

When rephrased to a suggestion that multi-sampling would be beneficial for weak samples this would make sense instead of describing that the results of this study support the added value of the approach (as one sample from each in this case would have generated identical results).

Author Response

We would like to thank the reviewers for their time and effort invested in our manuscript. We believe that their comments were an important part of the process and the corrections improved the manuscript significantly. Hopefully, we understood all the suggestions/comments correctly and made the appropriate corrections.

Again, thank you for all the suggestions.

Best,

Authors of the manuscript.

In the following, please, find the reviewer comments and our modifications:

Reviewer #1: The manuscript describes a succesful study of eight excavated skeletons from a mass grave. In general everything is described clearly and in detail. However, I'm rather surprised by the way the results are discussed / concluded. If I understand everything the right way, it seems like all the profiles of the pigmentation markers were complete and concordant. While this does support that the different bones came from the same skeleton it does not show much support for the multi-sample approach. 

Also, the mentioned 'consensus' profile that was used for calculation could just as well come from the same bone.

When rephrased to a suggestion that multi-sampling would be beneficial for weak samples this would make sense instead of describing that the results of this study support the added value of the approach (as one sample from each in this case would have generated identical results).

Thank you very much for your comments. Yes, all the profiles of the pigmentation markers were complete and concordant for all three bones that belonged to each individual skeleton, which indeed confirms the reliability of phenotypic prediction. It has to be noted that not only the multy-sample approach, but also increased number of PCR cycles (where less than 1 ng DNA input was available from bones) additionally contributed to those excelent results. In forensic genetics duplication analysis should be performed for obtaining reliable results.

The results of the single-test approach applied to bones up to 80 years old showed that only half of the samples analyzed produced a full HIrisPlex-S profile because it was not possible to use an optimal amount of template DNA (for the PCR-MPS HIrisPlex reaction 1 ng of template DNA is optimal) for phenotyping for all the samples due to low bone DNA quantity (Kukla-Bartoszek et al. 2020). Given that the authors of the study observed unsuccessful typing of a larger number of phenotypic SNPs in bone samples whose DNA amount does not reach a total of 1 ng template input for PCR-MPS phenotyping, it is difficult to expect that the repetition of analysis of the same sample would improve the phenotyping prediction. Moreover, even if an optimal amount of DNA input (1 ng) was used on Second World War petrous bones, different levels of AUC loss were computed in 4.2% of the HIrisPlex markers, showing that it is not possible to fully rule out allelic drop-out events even if an optimal amount of template DNA is used for phenotyping (Zupanič Pajnič et al. 2022) and repetition of phenotyping is necessary. In that case, we can expect that repetition of phenotyping of the same sample would help in increasing the reliability of phenotypic prediction. However, in Second World War skeletal remains we can not expect high quality DNA extracts and the most prommising approch for reliable phenotyping is sampling of multiple bones that according to published papers yield the highest amounts of DNA and perform phenotyping on more skeletal element types. When a multi-sample analysis method with consensus profile generation was performed on 11 skeletons dated from the 3rd to the 18th centuries AD using an MPS-based HIrisPlex system (at least 4 different skeletal element types were processed per each skeleton), consensus typing was achieved for 92% of the markers in 10 out of 11 ancient skeletons (Zupanič Pajnič et al. 2022), showing the possibility of improving the phenotyping prediction reliability through analysis of multiple skeletal element types. No full consensus profiles were obtained in any ancient skeleton analyzed. Because Second World War skeletal remains are better preserved, full consensus profiles can be expected and in study submitted, full profiles were obtained from all bones analyzed. It is necessary to note that increased number of PCR cycles contributed to successful typing of low template samples studied. Concordant full profiles proved the reliability of phenotyping of Second World War victims.

To additionaly emphasize the difference between different bone type, the following explanation was added under Introduction section: “Gray (1918) classifies human bones as short, long, flat, irregular, or sesamoid, and bones also vary by shape and function. Consequently, bone composition also varies because some bones have thicker tissue that is more compact with greater mineralization, and other bones are mostly comprised of trabecular bone covered by a thin to medium-thick layer of compact bone. Long bones are predominantly thick compact bone with a limited share of cancellous bone, whereas short, sesamoid, flat, and irregular bones primarily have cancellous bone in comparison to compact bone tissue (Gray 1918).

Reference Gray 1918 was added to the Reference list:

Gray H (1918) Gray’s anatomy of the human body, 20th ed

The following explanation was added under Discussion section: “A recent study performed on ancient skeletons using an MPS-based HIrisPlex system also proposed a multi-sample analysis method with consensus profile generation as a suitable method for eye and hair color prediction of ancient DNA obtained from 11 skeletons dated from the 3rd to the 18th centuries AD [15].”

Reviewer 2 Report

Dear Editor of Genes and Authors,

I have read the manuscript titled «A multi-sample approach in forensic phenotyping of aged skeletal remains using massive parallel sequencing (MPS) technology» and have found it well-written and explored, technically sound and offering a suitable literature review. It is also a relevant work for the journal Genes, and I only have two comments that I would like to be addressed:

1.       Authors allude several times to «old bones» or «aged bones» as a synonym to «not recent bones», but the wording can be confusing because it can be understood as the aging process of the skeleton. Thus, I would add something like «chronological old bones» or «ancient bones», as opposed to recent, forensic, bones.

2.       The authors provide a very good technical procedure for eye and hair color prediction through DNA analysis but fail to give a more thorough justification for its use in the identification of missing persons, especially in the context of mass victim identification. How can we reconcile the obtained data characteristics (for example, having brown eyes) in a context where the majority of people have such characteristics?

Best regards.

Author Response

We would like to thank the reviewers for their time and effort invested in our manuscript. We believe that their comments were an important part of the process and the corrections improved the manuscript significantly. Hopefully, we understood all the suggestions/comments correctly and made the appropriate corrections.

Again, thank you for all the suggestions.

Best,

Authors of the manuscript.

In the following, please, find the reviewer comments and our modifications:

Reviewer #2:

I have read the manuscript titled «A multi-sample approach in forensic phenotyping of aged skeletal remains using massive parallel sequencing (MPS) technology» and have found it well-written and explored, technically sound and offering a suitable literature review. It is also a relevant work for the journal Genes, and I only have two comments that I would like to be addressed:

  1. Authors allude several times to «old bones» or «aged bones» as a synonym to «not recent bones», but the wording can be confusing because it can be understood as the aging process of the skeleton. Thus, I would add something like «chronological old bones» or «ancient bones», as opposed to recent, forensic, bones.

Thank you very much for your comment. We totally agree with you, thus «old bones» or «aged bones» were replaced with «chronological old bones» throughout the manuscript.

  1. The authors provide a very good technical procedure for eye and hair color prediction through DNA analysis but fail to give a more thorough justification for its use in the identification of missing persons, especially in the context of mass victim identification. How can we reconcile the obtained data characteristics (for example, having brown eyes) in a context where the majority of people have such characteristics?

Thank you very much for your comment. We totally agree with you, thus following explanation was added under Discussion section: “The HIrisPlex system using SNaPshot technology followed by capillary electrophoresis (CE) has been applied to analysis of victims from Second World War mass graves in Slovenia (Chaitanya et al. 2017), and the results have effectively demonstrated the practical usefulness of predicting phenotypic traits. Namely, two femurs from the same mass grave were found to belong to brothers based on STR typing. With the help of phenotypic prediction and anamnestic data on their appearance, based on the testimony of their sister, it was determined which brother was which (Chaitanya et al. 2017). Similar was done by Zupanič Pajnič [19] by applying the HIrisPlex system and massive parallel sequencing (MPS) technology to a female victim of an elite prewar couple killed during the Second World War. The female victim from the grave could not be identified through autosomal STR and mitochondrial DNA analyses because she had no living relatives [19]. However, she was a prominent person of the time, and her portrait hangs in the Ljubljana City Museum, and so it was possible to compare the eye and hair color with her appearance in the portrait [19].”

Reference Chaitanya et al. 2017 was added to the Reference list:

Chaitanya L, Zupanič Pajnič I, Walsh S, Balažic J, Zupanc T, Kayser M. Bringing colour back after 70 years: predicting eye and hair colour from skeletal remains of World War II victims using the HIrisPlex system. Forensic Sci Int Genet. 2017;26:48–57.